# Engineering bacterial thiosulfate and tetrathionate sensors for detecting gut inflammation

Kristina N-M Daeffler[1] (ID), Jeffrey D Galley[2], Ravi U Sheth[1], Laura C Ortiz-Velez[2], Christopher O Bibb[3], Noah F Shroyer[4] (ID), Robert A Britton[2] & Jeffrey J Tabor[1,5,*] (ID)

## Abstract

There is a groundswell of interest in using genetically engineered sensor bacteria to study gut microbiota pathways, and diagnose or treat associated diseases. Here, we computationally identify the first biological thiosulfate sensor and an improved tetrathionate sensor, both two-component systems from marine *Shewanella* species, and validate them in laboratory *Escherichia coli*. Then, we port these sensors into a gut-adapted probiotic *E. coli* strain, and develop a method based upon oral gavage and flow cytometry of colon and fecal samples to demonstrate that colon inflammation (colitis) activates the thiosulfate sensor in mice harboring native gut microbiota. Our thiosulfate sensor may have applications in bacterial diagnostics or therapeutics. Finally, our approach can be replicated for a wide range of bacterial sensors and should thus enable a new class of minimally invasive studies of gut microbiota pathways.

**Keywords** diagnostic bacteria; gut inflammation; tetrathionate; thiosulfate; two-component system

**Subject Categories** Microbiology, Virology & Host Pathogen Interaction; Synthetic Biology & Biotechnology

**Mol Syst Biol. (2017) 13: 923**

## Introduction

The mammalian colon (gut) plays important roles in metabolism (Tremaroli & Backhed, 2012), and immune (Hooper *et al*, 2012) and brain function (Mayer *et al*, 2014). Gut processes are orchestrated by metabolic and signaling interactions between host cells and the dense and diverse community of resident bacteria (the microbiota). Disruptions in these interactions due to host genetics, environmental agents, or changes to the composition or physiological activity of the microbiota are linked to a spectrum of diseases including obesity (Ridaura *et al*, 2013), inflammation (Winter *et al*, 2013a), cancer

(Schulz *et al*, 2014), and depression (Foster & McVey Neufeld, 2013). However, due to the complexity and relative inaccessibility of the gut environment, and the challenges in constructing realistic *in vitro* gut models, these processes remain poorly understood.

Genetically engineered sensor bacteria have untapped potential as tools for analyzing gut pathways. Bacteria have evolved sensors of a large number of gut-relevant molecules. Such sensors could be repurposed and used to control the expression of reporter genes, enabling minimally invasive measurements of gut metabolites. In three previous studies, gut-adapted bacteria engineered to express colorimetric and luminescent reporter genes under the control of different chemically responsive transcriptional regulatory systems (sensors) were administered to mice by oral gavage and used to detect the corresponding chemicals in the gut via reporter assays of fecal samples (Drouault *et al*, 2002; Kotula *et al*, 2014; Mimee *et al*, 2015). However, the chemicals sensed in these studies—tetracycline, isopropyl β-D-1-thiogalactopyranoside (IPTG), and various sugars—are not produced within the gut environment or linked to gut pathways, but were administered to the animals via the diet. In a fourth study, a sensor bacterium was designed to measure host fucose levels in response to Toll-like receptor activation via fluorescence microscopy (Pickard *et al*, 2014). However, the mice in this study, and the three previous studies, were either raised in germ-free conditions or pre-treated with antibiotics to clear the native microbiota prior to administration of the engineered sensor strains. This sweeping perturbation has major effects on gut physiology, making these methods poorly suited to the analysis of gut pathways. In light of these studies, two major current challenges are to (i) engineer bacterial strains that sense other metabolites produced in the gut, and (ii) develop methods to assay reporter gene expression from those strains in animals with an intact microbiota.

Gut sulfur metabolism is linked to inflammation (colitis) via poorly understood microbe–host interactions. Sulfate-reducing bacteria (SRB) present in the colon produce hydrogen sulfide ($H_2S$) from oxidized sulfur species derived from the host and diet, a process that has been suggested to be involved in colitis (Roediger *et al*, 1997; Blachier *et al*, 2010). At high concentrations, $H_2S$ can be toxic to host cells due to its ability to outcompete $O_2$ for metal

1 Department of Bioengineering, Rice University, Houston, TX, USA
2 Department of Molecular Virology and Microbiology, Baylor College of Medicine, Houston, TX, USA
3 Department of Pathology, Texas Children's Hospital, Houston, TX, USA
4 Department of Medicine, Baylor College of Medicine, Houston, TX, USA
5 Department of Biosciences, Rice University, Houston, TX, USA
*Corresponding author. Tel: +1 713 348 8316; E-mail: jeff.tabor@rice.edu

cofactor binding in cytochrome *c* oxidase, and thereby prevent oxidative phosphorylation (Petersen, 1977; Nicholls & Kim, 1982; Khan *et al*, 1990). Additionally, $H_2S$ inhibits butyrate oxidation by colonic epithelial cells *in vitro* (Roediger *et al*, 1993; Moore *et al*, 1997), the preferred method of energy production by these cells (Roediger, 1982), which has also been observed in biopsies of patients with ulcerative colitis (Chapman *et al*, 1994). However, attempts to directly link $H_2S$ to inflammation have generated conflicting results due to difficulties in measuring $H_2S$ within complex biological samples and challenges in using chemical donors to recreate physiological $H_2S$ levels *in vitro* (Pitcher *et al*, 2000; Huycke & Gaskins, 2004; Nagy *et al*, 2014).

Thiosulfate ($S_2O_3^{2-}$) and tetrathionate ($S_4O_6^{2-}$) are appealing targets for studying the link between gut sulfur metabolism and inflammation. Host enzymes detoxify $H_2S$ to thiosulfate (Levitt *et al*, 1999; Jackson *et al*, 2012; Vitvitsky *et al*, 2015). Although enterobacteria and SRB can utilize thiosulfate as a terminal electron acceptor (TEA) in anaerobic respiration, the reaction is energetically unfavorable and unlikely to occur in the gut due to the availability of more desirable substrates (Barrett & Clark, 1987; Stoffels *et al*, 2012). Furthermore, using a *Salmonella typhimurium* mouse model, Winter and colleagues have shown that reactive oxygen species (ROS) produced by the host during inflammation convert thiosulfate to tetrathionate, which this pathogen consumes to establish a foothold for infection (Winter *et al*, 2010). Thus, colonic thiosulfate and tetrathionate levels may correlate with pro-inflammatory conditions. However, thiosulfate has not been evaluated as an inflammation biomarker and tetrathionate has not been studied in other inflammation models.

Here, we set out to engineer gut bacteria to sense and report thiosulfate and tetrathionate levels in the widely used dextran sodium sulfate (DSS) mouse model of colitis. However, there is no known genetically encoded thiosulfate sensor and the only known genetically encoded tetrathionate sensor is the TtrSR two-component system (TCS) from *S. typhimurium* (Hensel *et al*, 1999; Price-Carter *et al*, 2001). This TCS comprises TtrS, a membrane-bound sensor histidine kinase (SK) that phosphorylates the cytoplasmic response regulator (RR) TtrR in the presence of tetrathionate. Phosphorylated TtrR (TtrR~P) activates transcription of the tetrathionate reductase operon, *ttrBCA*, via the *ttrB* promoter ($P_{ttrB}$). However, $P_{ttrB}$ is repressed by $O_2$ and nitrate via the global regulator FNR and an unknown pathway, respectively (Price-Carter *et al*, 2001). Furthermore, FNR is required for transcription from $P_{ttrB}$ (Price-Carter *et al*, 2001), eliminating the possibility of avoiding $O_2$ cross-repression by deleting this repressor. Though gut $O_2$ levels are incompletely understood and an area of active study, they may be relatively high near the epithelial mucosal boundary due to proximity to the blood. Furthermore, gut nitrate levels have been shown to be elevated during inflammation (Winter *et al*, 2013b). Thus, the unwanted cross-regulation of *S. typhimurium* TtrSR could comprise its performance as a gut tetrathionate sensor.

In this study, we computationally identify a novel TCS from the marine bacterium *Shewanella halifaxensis* HAW-EB4 and characterize it in laboratory *Escherichia coli*, demonstrating that it is the first known biological thiosulfate sensor. We similarly identify a TtrSR homolog from the marine bacterium *Shewanella baltica* OS195 that is only weakly repressed by $O_2$ and not repressed by nitrate in *E. coli*. We optimize the performance of both sensors in the

probiotic strain *E. coli* Nissle 1917 and gavage these engineered strains into mice without antibiotic pre-treatment. We then use flow cytometry to detect the engineered bacteria among the native gut microbiota in colon and fecal samples and quantify sensor outputs. Using histologic scoring, we demonstrate that our thiosulfate sensor is activated by colon inflammation, suggesting thiosulfate may be a novel biomarker and that our sensor bacteria have potential as a non-invasive diagnostic of colitis. Our tetrathionate sensor has low *in vivo* activity even at high inflammation levels, suggesting this molecule may not be produced in the DSS model or that it is rapidly degraded by the gut microbiota.

# Results

## Bioinformatic identification of candidate thiosulfate- and tetrathionate-sensing TCSs

*Salmonella typhimurium* TtrS likely binds tetrathionate via a periplasmic sensing domain with similarity to the *E. coli* phosphonate-binding protein PhnD (Appendix Fig S1A). PhnD is involved in active transport of alkylphosphonates across the inner membrane (Metcalf & Wanner, 1993). Thiosulfate is chemically similar to alkylphosphonates and tetrathionate (i.e., a −2 charge, three oxygens around a central atom, and a similar molecular geometry), and could be sensed by a similar ligand-binding domain. Additionally, *ttrSR* resides adjacent to a three-gene cluster encoding a tetrathionate reductase in the genome (Hensel *et al*, 1999) (Appendix Fig S1B). Thus, we hypothesized that a bioinformatic search for a TCS containing a sensor kinase (SK) with a PhnD-like sensor domain located near a thiosulfate reductase might reveal an uncharacterized thiosulfate sensor.

We searched the UniProtKB sequence database for all SKs with PhnD-like sensor domains, resulting in 838 proteins (Materials and Methods and Dataset EV1). Then, we enriched for unique SKs by eliminating those > 70% identical in sequence to any other protein in the list, yielding 154 candidates. One hundred and thirty-one of these SKs reside adjacent to an RR in their native genomic context, indicating a likely signaling interaction between the two proteins. Of these putative TCSs, 13 reside adjacent to predicted thiosulfate utilization genes, while 18 reside adjacent to predicted tetrathionate utilization genes (Dataset EV1).

Based on three lines of reasoning, we selected Shal_3128/9 from *S. halifaxensis* HAW-EB4 (hereafter *S. halifaxensis*) and Sbal195_3859/8 from *S. baltica* OS195 (hereafter *S. baltica*), as candidate thiosulfate and tetrathionate sensors for further analysis. First, *Shewanella* sp. couple energy production to the reduction of a wide range of TEAs including thiosulfate (Burns & DiChristina, 2009) and tetrathionate (Myers & Nealson, 1988; Qiu *et al*, 2013). Second, *Shewanella* and *E. coli* are both γ-proteobacteria, increasing the likelihood that TCSs can be successfully ported between them. Third, a majority of known *Shewanella* reductase promoters are not regulated by FNR or nitrate (Maier & Myers, 2001; Wu *et al*, 2015), reducing the chance of unwanted cross-regulation in *E. coli*.

## Shal_3128/9 is a thiosulfate sensor (ThsSR)

The candidate thiosulfate-sensing SK Shal_3128 and predicted FixJ-like RR Shal_3129 reside adjacent to a 342-bp intergenic region

(hereafter $P_{phsA342}$; Appendix Fig S2A) upstream of Shal_3127-5, which encodes a predicted thiosulfate reductase (Fig 1A). We hypothesized that this intergenic region contains a Shal_3129-activated promoter. Previously, we have shown that RR overexpression in the absence of the cognate SK and input can strongly activate the output promoter (Schmidl *et al*, 2014), possibly due to RR phosphorylation by alternative sources (small molecules, non-cognate SKs), or low-affinity binding by non-phosphorylated RRs. Thus, we constructed plasmid pKD184 (Appendix Fig S3A) wherein an *E. coli* codon-optimized Shal_3129 gene is expressed under control of an anhydrotetracycline (aTc)-inducible promoter, and $P_{phsA342}$ resides upstream of the fluorescent reporter gene superfolder GFP (*sfgfp*). We then grew a laboratory *E. coli* strain (BW28357) carrying pKD184 under increasing aTc concentrations and measured the corresponding sfGFP levels by flow cytometry (Materials and Methods). Indeed, sfGFP fluorescence increases from 810 ± 210 to 6,380 ± 280 Molecules Equivalent Fluorescein (MEFL) (Materials and Methods) over this range (Appendix Fig S4). Furthermore, mutation of the Shal_3129 phosphoryl-accepting aspartate residue to a non-functional alanine (D57A) attenuates this response (Appendix Fig S4). We conclude that Shal_3129 encodes a RR that activates transcription from $P_{phsA342}$ in a phosphorylation-dependent manner.

Next, we constructed pKD182 (Appendix Fig S3B), containing an *E. coli* codon-optimized Shal_3128 gene under control of an IPTG-inducible promoter, and co-transformed it into the strain containing pKD184 (Fig 2A). We grew the bacteria in 0 and 5 mM thiosulfate at different aTc and IPTG concentrations and analyzed sfGFP as before. In the absence of thiosulfate, Shal_3129

induction again activates sfGFP expression, but this activation is reduced by induction of Shal_3128 (Appendix Fig S5A). Thiosulfate increases sfGFP in a manner strongly dependent upon Shal_3129 and Shal_3128 expression (Appendix Fig S5B) with an optimal dynamic range (ratio of sfGFP in the presence versus absence of thiosulfate) of 21 ± 2 fold (0 mM thiosulfate, 670 ± 100 MEFL; 5 mM thiosulfate, 13,600 ± 2,080 MEFL) (Fig 2B and Appendix Fig S5C).

To validate that the observed responses are due to canonical TCS signaling rather than an alternative pathway, we independently introduced four perturbations to Shal_3128/9. Specifically, we mutated the conserved catalytic histidine (Shal_3128 H372) and phosphoryl-accepting aspartate (Shal_3129 D57) to non-functional alanines, eliminated the Shal_3128 expression plasmid, and deleted the Shal_3129 DNA binding domain. Each of these perturbations abolishes thiosulfate activation (Fig 2B). Taken together, these results indicate that Shal_3128 encodes a bifunctional SK that dephosphorylates and phosphorylates Shal_3129 in the absence and presence of thiosulfate, respectively. Accordingly, we renamed Shal_3129 and Shal_3128 *thsR* and *thsS* for thiosulfate response regulator and sensor, respectively.

### ThsSR sensitivity and specificity

Sensitivity and specificity are desirable properties of engineered sensors. To examine sensitivity, we characterized the dose–response relationship, or transfer function, of ThsSR for thiosulfate. ThsSR output increases in a manner well fit by an activating Hill function with half-maximal activation ($k_{1/2}$) at 280 ± 10 μM and Hill coefficient ($n$) 1.8 ± 0.1 (Fig 2C). To examine specificity, we exposed the ThsSR-expressing strain to a panel of eight alternative TEAs that *Shewanella* use for anaerobic respiration and that may be present in the gut (sulfate, sulfite, tetrathionate, DMSO, nitrate, nitrite, TMAO, and fumarate) at a concentration well above what is expected in the gut (10 mM). ThsSR does not respond to any of these alternative ligands (Fig 2D) several of which are chemically similar, thus demonstrating high specificity.

Thiosulfate has poor energy generating potential, and in some cases, facultative anaerobes repress reductases for less preferred substrates in the presence of more desirable substrates (Gunsalus, 1992). Thus, we hypothesized that ThsSR might be repressed by more favorable TEAs used in anaerobic respiration. To test this hypothesis, we simultaneously exposed the ThsSR-expressing strain to 5 mM thiosulfate and 10 mM of each of the eight alternative TEAs (Appendix Fig S6), all of which have higher energy generating potential than thiosulfate. While six have no effect, sulfite and tetrathionate inhibit thiosulfate activation (Appendix Fig S6A). We analyzed the corresponding "repression transfer functions", which reveal that these ligands inhibit ThsSR by up to 94% and 87%, with half-maximal inhibition at 390 ± 10 μM and 550 ± 20 μM, respectively (Appendix Fig S6B). We performed Schild plot analysis to evaluate the mechanism of inhibition, but the data are inconclusive (Appendix Fig S7).

### Identification of $P_{phsA342}$ regulatory elements

$P_{phsA342}$ contains a predicted promoter downstream of 18-bp direct repeat sequences separated by a consensus cAMP repressor protein

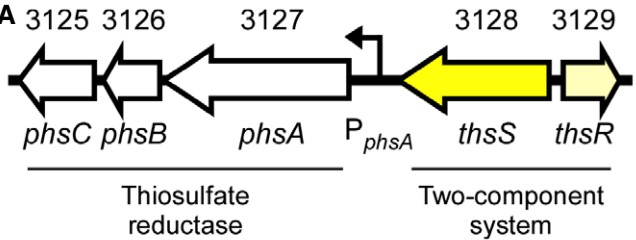

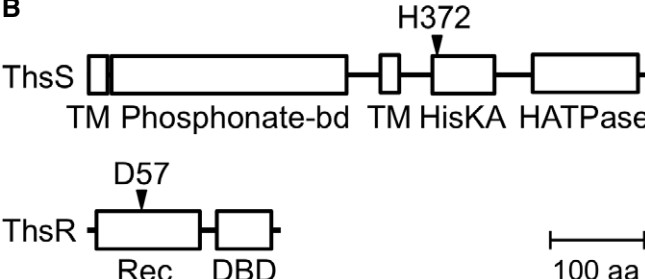

**Figure 1. ThsS/ThsR gene locus and domain layout.**

A The *Shewanella halifaxensis* genomic region containing the thiosulfate reductase operon, PhsAB and PsrC (Shal_3125-7), neighboring thiosulfate-sensing TCS, ThsS/R (Shal_3128/9), and the ThsR activated promoter (PphsA).

B Predicted domain architecture of ThsS and ThsR. Residues involved in phosphotransfer are indicated with arrows.

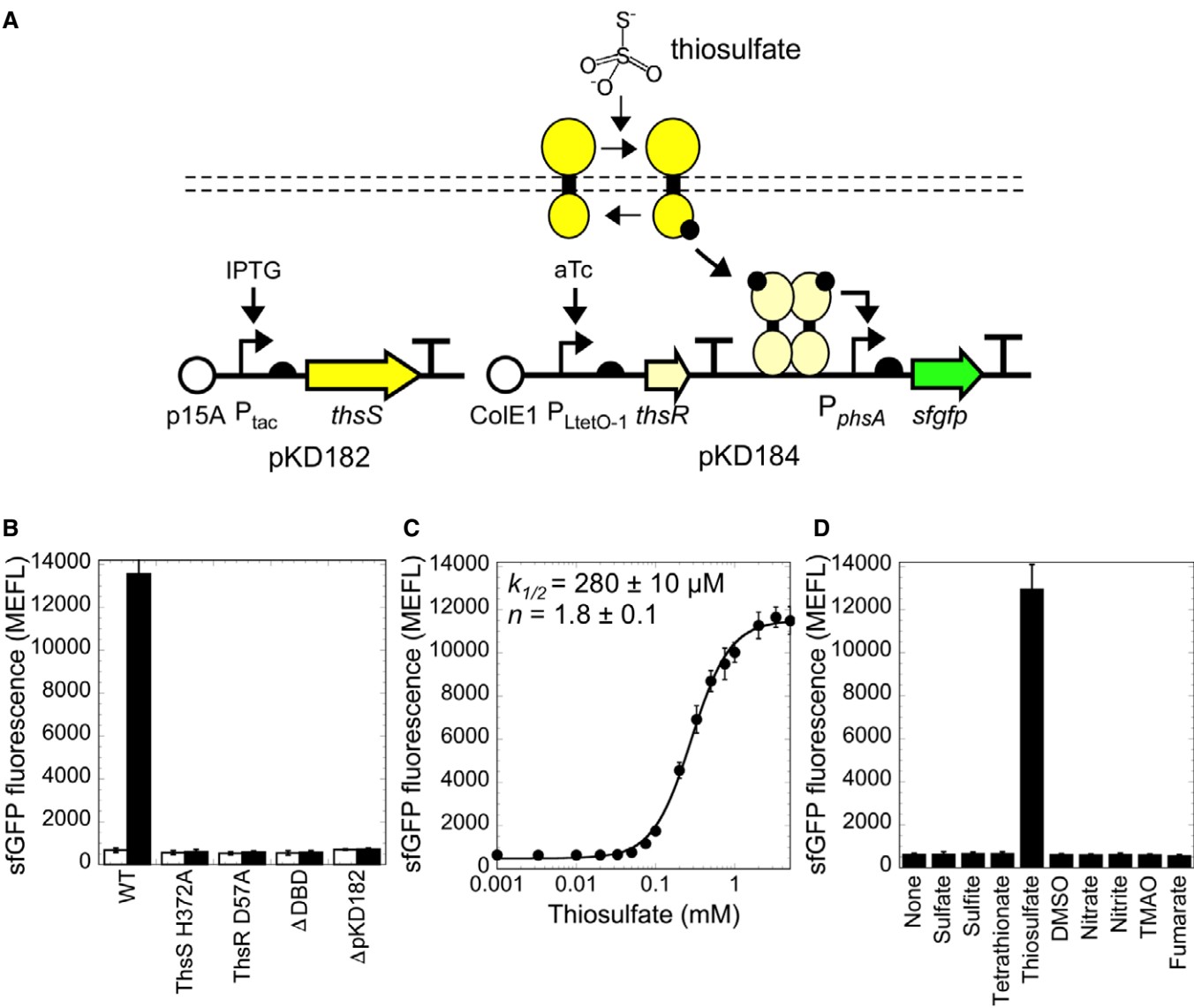

**Figure 2. Characterization of the thiosulfate sensor ThsSR.**

A   Schematic of ligand-induced signaling through ThsS/R and plasmid design of the aTc- and IPTG-inducible sensor components.
B   Ligand response in wild-type and inactivated mutant sensors. White bars are with no thiosulfate and black bars with 5 mM thiosulfate.
C   Thiosulfate dose–response curve.
D   Selectivity of ThsS/R to thiosulfate over other terminal electron acceptors. All ligands were tested at 10 mM concentration.

Data information: Data are mean of at least three biological replicates ± SD.

(CRP) operator (Appendix Fig S2). Each 18-bp element contains a 6-bp inverted repeat (TA<u>TGTGGTT</u>TT<u>ACCACAA</u>T), resembling a known FixJ operator site (Kurashima-Ito *et al*, 2005). Using a series of 5′ truncations, we determined that the first 151 bp, including the first 18-bp element, are dispensable (Appendix Fig S2B). On the other hand, truncation of the promoter through the CRP operator or mutagenesis of the CRP binding motif reduces transcriptional output and thiosulfate response (Appendix Fig S2B). Furthermore, truncation of the promoter through the second 18-bp element, or mutation of either 6-bp inverted repeat, abolishes thiosulfate activation (Appendix Fig S2B). These results indicate that the second 18-bp element is the primary ThsR

operator and that the CRP site plays a role in thiosulfate activation.

To further evaluate the role of CRP, we next examined whether ThsSR is affected by glucose. ThsSR is activated by thiosulfate in the presence of glucose, but absolute transcriptional output (0 mM thiosulfate, 280 ± 60 MEFL; 5 mM thiosulfate, 1,390 ± 340 MEFL) and activation (4.9 ± 0.2 fold) are reduced, similar to the levels observed when disrupting the CRP site in the absence of glucose (Appendix Fig S8). These data suggest that CRP binding is required for full activation of P*phsA342*, consistent with known anaerobic respiration pathways in *Shewanella* (Saffarini *et al*, 2003; Wu *et al*, 2015).

## Characterization and optimization of the tetrathionate sensor Sbal195_3859/8 (*Shewanella baltica* TtrSR)

We next characterized the putative tetrathionate sensor that we had identified computationally. The Sbal_3859/8 operon, encoding a *ttrSR* homolog, is separated from the predicted tetrathionate reductase operon *ttrBCA* by a 344-bp intergenic region (hereafter P*ttrB344*; Fig 3A and Appendix Fig S9). As before, we cloned aTc-inducible RR (Sbal195_3858, hereafter *S. baltica* TtrR, pKD226) and IPTG-inducible SK (Sbal195_3859 hereafter *S. baltica* TtrS, pKD227) plasmids (Appendix Fig S10) and validated that P*ttrB344* is a Sbal195_3858-activated promoter by overexpressing the RR (Appendix Fig S11). We next demonstrated that 1 mM tetrathionate results in a $30 \pm 10$ fold increase of sfGFP levels at the best

expression level (from $23 \pm 2$ MEFL to $680 \pm 250$ MEFL) (Fig 3D), indicating *S. baltica* TtrSR is indeed a tetrathionate sensor. In the absence of tetrathionate, increased expression of *S. baltica* TtrS decreases *S. baltica* TtrR-induced promoter activation, suggesting the former can also function as a phosphatase to modulate signaling (Appendix Fig S12).

P*ttrB344* contains a near-consensus FNR binding site and numerous repeat elements that could serve as operator sites (Appendix Fig S9A). To eliminate unnecessary and possibly detrimental sequence elements, we first screened a library of 5′ and 3′ P*ttrB344* truncations. We identified an 85-bp minimal promoter (P*ttrB185-269*) with reduced leakiness and markedly improved expression range (0 mM tetrathionate, $83 \pm 4$ MEFL; 1 mM tetrathionate, $3,730 \pm 470$ MEFL) relative to the full-length intergenic region (Fig 3D). We

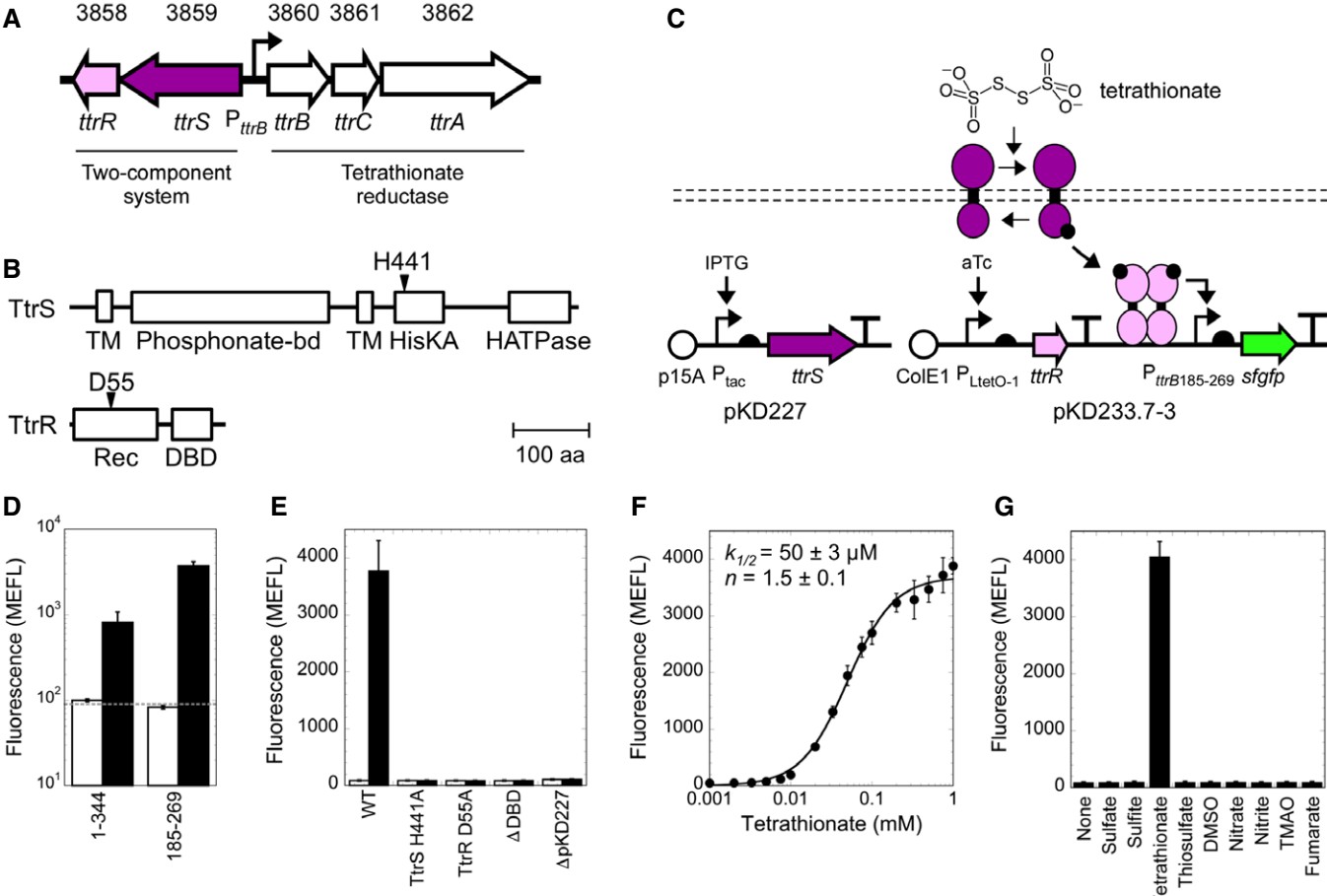

**Figure 3.  Characterization of the tetrathionate-sensing TCS, TtrS/R (Sbal195_3859/8).**

A   Location of the thiosulfate sensor, consisting of TtrS (Sbal195_3859) and TtrR (Sbal195_3858), and tetrathionate reductase, TtrBCA (Sbal195_3860-2), on the chromosome of *Shewanella baltica* OS195.

B   Predicted domain architecture of TtrS and TtrR with the phosphotransfer residues indicated with an arrow. Domains are labeled by their Pfam family names. A scale bar is included for reference.

C   Schematic of tetrathionate-induced activation and plasmid design of the aTc- and IPTG-inducible TtrSR components.

D   Tetrathionate-induced sfGFP production of the optimized truncated promoter P*ttrB185-269* compared to the full-length intergenic region P*ttrB344* in the presence (black bars) and absence (white bars) of 1 mM tetrathionate. The dotted horizontal line indicates cellular autofluorescence.

E   Tetrathionate-induced sfGFP production in the presence (black bars) and absence of 1 mM tetrathionate of wild-type and inactivated sensors.

F   Tetrathionate dose–response of the optimized promoter (closed circles).

G   Selectivity of TtrS/R to tetrathionate over other terminal electron acceptors. All ligands were tested at 10 mM concentration.

Data information: Data are mean of at least three biological replicates ± SD.

therefore used this truncated promoter for all subsequent experiments. Hereafter, we do not subtract *E. coli* autofluorescence from *S. baltica* TtrSR data as it is indistinguishable from the low sfGFP expression levels observed from this promoter in the absence of tetrathionate. Though most DNA upstream of the putative FNR operator can be deleted with minimal effect, truncation into this site reduces tetrathionate activation (Appendix Fig S9B), suggesting FNR plays a role in *S. baltica* TtrSR activation. Within $P_{ttrB185-269}$, we identified three FixJ-like inverted repeats that may be *S. baltica* TtrR binding sites (187-<u>ATTTG</u>NNNNNNNNNNNN<u>CAAAT</u>-207, 207-<u>TCCAC</u>NNNNNNNNNNN<u>GTGGA</u>-225, and 254-<u>TTTACAG</u>NNNNN<u>CTGTAAA</u>-272) (Appendix Fig S9C). However, our truncation and mutagenesis studies to identify the TtrR operator site are inconclusive (Appendix Fig S9D and E).

cAMP repressor protein has previously been shown to recognize and activate an FNR binding site (Sawers *et al*, 1997), suggesting the potential for glucose regulation of $P_{ttrB344}$. To explore this possibility, we tested the sensitivity of bacteria expressing *S. baltica* TtrSR to glucose. Though glucose decreases absolute transcriptional output, bacteria expressing this sensor are still activated by tetrathionate (0 mM tetrathionate, $81 \pm 6$ MEFL; 1 mM tetrathionate, $425 \pm 120$ MEFL) (Appendix Fig S13).

Using the same set of inactivating mutations as before, we validated that the tetrathionate response is due to canonical TCS phospho-signaling through *S. baltica* TtrSR (Fig 3E). We also measured the transfer function, revealing that *S. baltica* TtrSR responds to tetrathionate in a sigmoidal manner with greater sensitivity than ThsSR for thiosulfate ($k_{1/2} = 50 \pm 3$ μM) but a similar Hill coefficient ($n = 1.5 \pm 0.1$; Fig 3F). Using the alternative TEA panel, we demonstrated that *S. baltica* TtrSR is highly specific to tetrathionate (Fig 3G). Finally, unlike ThsSR, *S. baltica* TtrSR is not inhibited by any of the alternative TEAs (Appendix Fig S14).

## Optimizing ThsSR and *Shewanella baltica* TtrSR for the gut environment

We performed the above sensor development studies aerobically, in monoculture, in a domesticated laboratory strain, and using chemical inducers to optimize SK and RR expression levels. However, the mammalian gut has relatively little oxygen, contains a dense and diverse microbiota, is inhospitable to domesticated strains, and is not amenable to the use of chemical inducers. Therefore, we set out to adapt our sensors to the gut environment.

First, to eliminate the use of chemical inducers, we synthesized two paired plasmid libraries wherein the SK and RR for each sensor are expressed to different levels from constitutive promoters and ribosome binding sites (RBSs) of varying strengths (Appendix Figs S15 and S16). Then, we combinatorially transformed each paired plasmid library into the human probiotic strain *E. coli* Nissle 1917 (hereafter Nissle), and measured activation by the cognate ligand in aerobic conditions. The best ThsSR and *S. baltica* TtrSR plasmid combinations result in $7 \pm 2$ fold activation (0 mM thiosulfate, $290 \pm 30$ MEFL; 5 mM thiosulfate, $2,050 \pm 500$ MEFL) (Appendix Fig S15) and $37 \pm 7$ fold activation (0 mM tetrathionate, $87 \pm 9$ MEFL; 1 mM tetrathionate $3,220 \pm 630$ MEFL) (Appendix Fig S16), respectively.

To enable detection of our sensor bacteria among the native microbiota, we added a strong mCherry expression cassette to each

optimized RR plasmid. We then re-measured ligand activation, which revealed that this alteration does not change the performance of either sensor (Appendix Figs S15 and S16).

Then, we analyzed the performance of each Nissle sensor strain in anaerobic conditions *in vitro* (Materials and Methods). Interestingly, the ThsSR Nissle strain (Fig 4A and Appendix Fig S17) exhibits $430 \pm 30$ MEFL and $19,200 \pm 5,200$ MEFL in the absence and presence of thiosulfate in these conditions ($45 \pm 13$ fold activation), a sixfold improvement relative to aerobic conditions, even without subtracting Nissle autofluorescence which should increase the dynamic range estimate (Appendix Fig S15). However, anaerobic growth reduces the dynamic range of *S. baltica* TtrSR in Nissle due to elevated sfGFP in the absence of tetrathionate (Appendix Fig S16). This unwanted effect likely results from elevated TtrR concentrations. To recover the desired dynamic range, we screened a second plasmid library wherein *S. baltica* TtrR was expressed from weaker RBSs. We identified a variant (Fig 4D and Appendix Fig S18) with very low sfGFP ($238 \pm 24$ MEFL) in the absence of tetrathionate and high sfGFP ($19,800 \pm 670$ MEFL) in its presence ($84 \pm 9$ fold increase, with no autofluorescence subtraction) (Appendix Fig S16). Overall, our efforts yielded gut-optimized sensors with similar low and high outputs (and dynamic range) as the initial *in vitro*-optimized versions (Fig 4B and E). Finally, anaerobic growth results in a 17-fold (Fig 4F) increase in sensitivity of the re-optimized *S. baltica* TtrSR toward tetrathionate, possibly due to a reduction in FNR binding to $P_{ttrB185-269}$ relative to aerobic conditions.

To examine whether our sensor bacteria function in the complex colonic environment, whole colons were excised from healthy mice, tied, and injected with sensor bacteria and either 0 mM or 5 mM thiosulfate or 0 mM or 1 mM tetrathionate. After 6 h of incubation in DMEM, we collected the colon contents, homogenized the samples, filtered them to remove large particles, treated them with a translational inhibitor, and incubated them aerobically to allow sfGFP and mCherry to mature (Materials and Methods and Appendix Fig S19). Finally, we analyzed sfGFP expression by flow cytometry, using mCherry expression to identify our sensor bacteria among the native microbiota and other particles (Appendix Fig S20).

Each ligand activates its corresponding sensor, and these responses are attenuated when the TCSs are inactivated by mutation (Appendix Figs S21 and S22). Absolute sfGFP levels in the ligand activated state are attenuated in the colons relative to the *in vitro* experiments, especially for ThsSR. Colons injected with thiosulfate or tetrathionate smelled strongly of sulfide after 6 h of incubation, indicating bacterial reduction of the sulfur-containing metabolites and the potential for inhibition of ThsSR by metabolically produced sulfite. Additionally, the high levels of glucose present in DMEM may have partially inhibited promoter output, consistent with our *in vitro* experiments. Nonetheless, these results indicate that our sensors function in the complex colon environment.

## ThsSR is activated by gut inflammation

Next, we used our gut-optimized sensor strains to detect thiosulfate and tetrathionate in healthy and diseased mice (Fig 5A and Materials and Methods). We induced inflammation with DSS, one of the

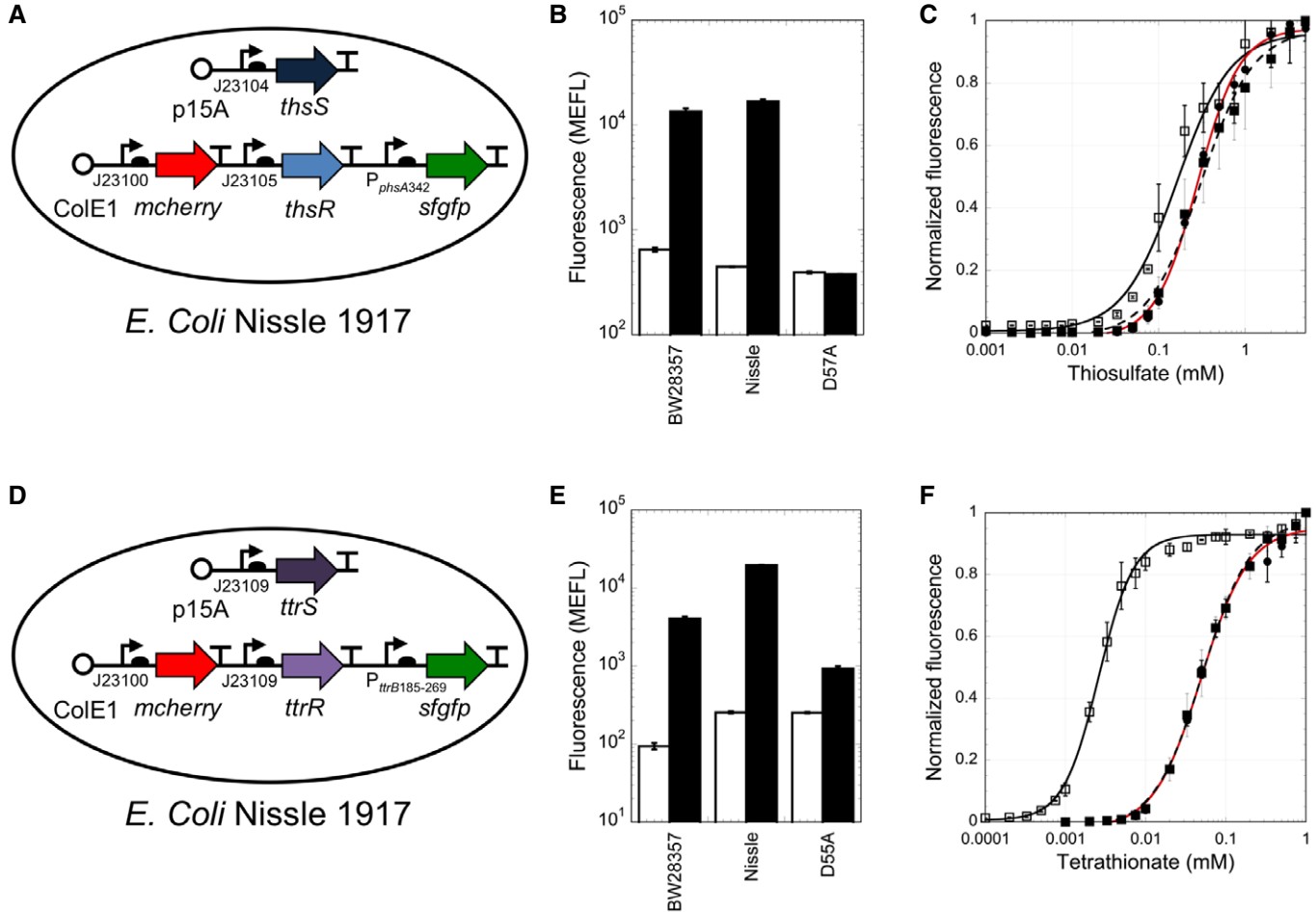

**Figure 4. Sensor optimization for thiosulfate and tetrathionate detection in the gut.**

A–F (A and D) Plasmid design of the constitutive sensors in *Escherichia coli* Nissle 1917. (B and E) Comparison of the inducible sensors in BW28357, the constitutive sensors in Nissle 1917, and D-to-A inactivated sensors for thiosulfate (B) and tetrathionate (E). GFP output is shown in the absence (white bars) and presence of 1 mM tetrathionate or 5 mM thiosulfate (black bars), respectively. (C and F) Normalized dose–response relationship of thiosulfate (C) and tetrathionate sensors (F). Shown is the original inducible BW28357 strain grown aerobically (closed circles, red curve fit), and a constitutive promoter strain in Nissle grown aerobically (closed squares) or anaerobically (open squares). Different constitutive promoters were used for the aerobic and anaerobic Nissle strains to achieve the best dynamic range. A shift in half-maximal response indicates sensitivity to oxygen.

Data information: Data are mean of at least three biological replicates ± SD.

most commonly used models for studying colitis (Chassaing *et al*, 2014). *In vitro* control experiments demonstrate that neither ThsSR nor *S. baltica* TtrSR responds to DSS or its sulfate moiety, and the presence of DSS does not inhibit sensor performance (Appendix Fig S23). Mice were administered either control drinking water or drinking water plus 3% DSS for 5 days. On day five, we orally gavaged the control and DSS-treated groups with $10^9$ bacteria expressing either ThsSR ($n = 14$), *S. baltica* TtrSR ($n = 8$), or one of the negative controls [ThsSR (D57A) ($n = 14$) or *S. baltica* TtrSR (D55A) ($n = 8$)]. Six hours later, we collected fecal, distal colon, and proximal colon samples.

sfGFP expression from the ThsSR strain is significantly higher in fecal, distal colon, and proximal colon samples of DSS-treated mice relative to healthy controls ($P < 0.01$) (Fig 5B, and Appendix Figs S24 and S25), while that from the ThsSR (D57A) strain is consistently low in both healthy and DSS-treated mice (Fig 5C). These results demonstrate that ThsSR can be activated in a living mouse

gut and indicate that thiosulfate may be elevated upon DSS treatment. Additionally, sfGFP levels measured in fecal samples are very similar to those measured in both the proximal and distal colon samples, suggesting that our fecal sampling method can be used to non-invasively analyze *in vivo* metabolite levels.

Next, we used histologic scoring to quantify inflammation levels in the colon of each mouse gavaged with ThsSR and ThsSR (D57A). Briefly, two blinded histopathologists assigned a value to the extent of epithelial damage and inflammatory infiltration in the mucosa, submucosa, and muscularis/serosa, resulting in an overall score from 0 (no inflammation) to 36 (maximal inflammation) (Chassaing *et al*, 2014). Water-treated animals exhibited low inflammation while DSS-treated animals had elevated inflammation with areas of focal ulceration (Appendix Fig S26A). We observe a weak correlation between fluorescence output and histopathology score for the wild-type sensor but not the inactive D57A sensor (Appendix Fig S27). Notably, four of the DSS-treated mice showed no ThsSR

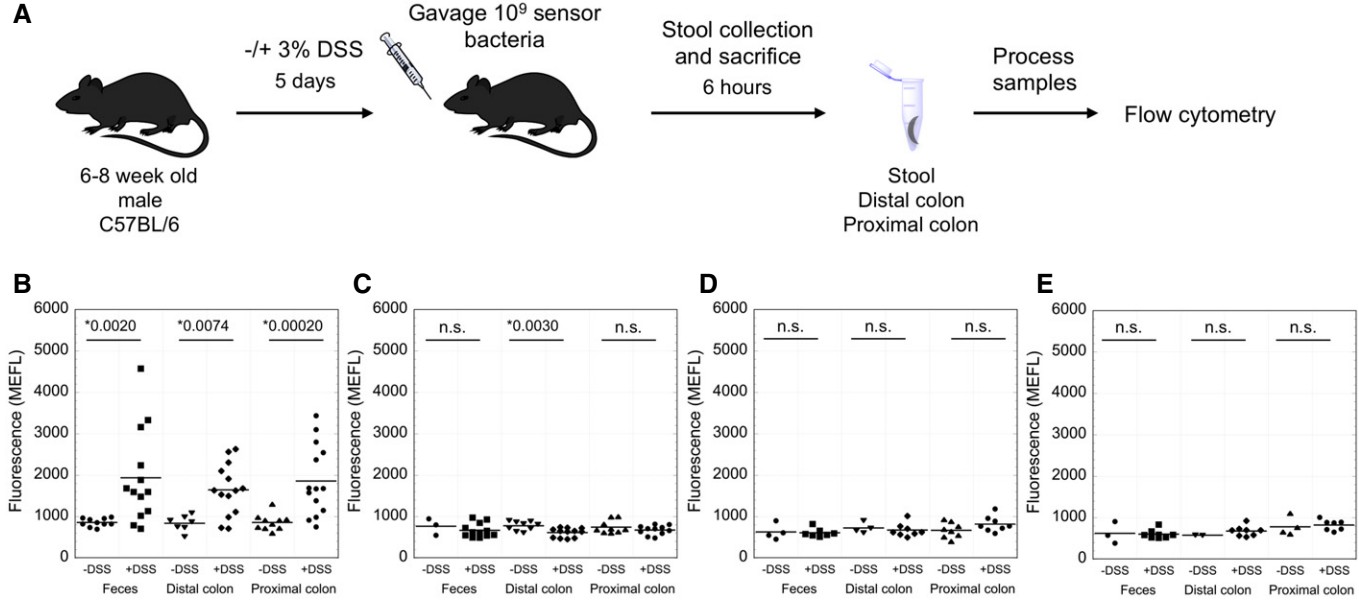

**Figure 5. *In vivo* measurement of thiosulfate and tetrathionate in healthy and inflamed mice.**

A    Experimental design. 6- to 8-week-old C57BL/6 mice were given water with or without 3% DSS for 5 days before oral gavage with sensor bacteria. After 6 h, samples were collected from the mice, processed, and analyzed by flow cytometry to measure GFP production.

B–E    Mice were gavaged with $10^9$ bacteria of the (B) thiosulfate sensor (*n* = 14), (C) inactivated thiosulfate sensor (D57A) (*n* = 14), (D) tetrathionate sensor (*n* = 8), or (E) inactivated tetrathionate sensor (D55A) (*n* = 8). Horizontal lines are the mean fluorescence. Asterisks indicate *P* < 0.05 with the *P*-value indicated, n.s. is indicated when *P* > 0.05. *P*-values were calculated using the *t*-test.

activation in any tissue tested despite elevated colonic inflammation. Three of these mice were the sole occupants of a single cage, suggesting cage-level variability in ThsSR function. Finally, we evaluated the diagnostic performance of the ThsSR sensor to predict DSS treatment by generating a receiver operating characteristic (ROC) curve (Appendix Fig S28). The area under the curve (AuROC) is 0.8692, reflecting a low false-positive rate.

*Shewanella baltica* TtrSR output is consistently low, and similar to *S. baltica* TtrSR (D55A), in all mice (Fig 5D and E). Mice treated with DSS had elevated inflammation relative to healthy controls and similar histologic scores as the DSS-treated mice given the ThsSR strain (Appendix Fig S26B). These results suggest that either our engineered *S. baltica* TtrSR construct does not function *in vivo*, or tetrathionate concentration in the lumen of DSS-treated mice, where our sensor bacteria likely reside in this 6-h protocol, is below the ~1 μM limit of detection (Fig 4F).

## Discussion

We have discovered and validated the first genetically encoded thiosulfate sensor (ThsSR), and a tetrathionate sensor (*S. baltica* TtrSR) with improved performance features relative to the only previously known variant. Both sensors are TCSs likely involved in anaerobic respiration in marine *Shewanella* sp. Unlike previously characterized reductase promoters from *E. coli* and other facultative anaerobes [e.g., TMAO (Iuchi & Lin, 1987), fumarate (Jones & Gunsalus, 1987), and DMSO (Cotter & Gunsalus, 1989)], both of our sensors are free from nitrate cross-repression and function in the presence and absence of oxygen. These benefits stem from the differences in

the *Shewanella* respiration regulatory network relative to other facultative anaerobes, whereby gene expression of anaerobic reductases is coordinated by CRP rather than the oxygen regulator FNR or the redox regulator ArcBA (Saffarini *et al*, 2003; Wu *et al*, 2015).

We do not anticipate that the inhibitory effects of glucose or sulfite will impact ThsSR for our purposes, because the concentration of both of these molecules is expected to be low in the colon (Wilson, 1962; Mishanina *et al*, 2015). Indeed, glucose repression could be exploited as the absence of glucose could serve as a signal that sensor bacteria are in the colon rather than in *in vitro* growth media or further upstream in the gastrointestinal tract. However, the elevated tetrathionate levels previously observed in *S. typhimurium* inflammation could repress ThsSR, leading to false-negative readouts. Thus, tetrathionate must be measured to ensure faithful thiosulfate reporting by ThsSR. Because ThsSR is activated, and *S. baltica* TtrSR activity is very low, we suspect there is no appreciable tetrathionate in our *in vivo* experiments.

Our flow cytometry-based method enables reliable measurement of the sfGFP expression levels of engineered sensor bacteria residing within complex colon and fecal samples. This method has several benefits compared to existing alternatives. First, unlike a previous approach involving luciferase measurements of bulk fecal samples (Mimee *et al*, 2015), flow cytometry enables measurement of bacterial populations at single cell resolution, providing far more information about the true response of the sensor and potentially the gut environment. Additionally, our protocol does not require the use of digital-like genetic memory circuits, which were used in two previous studies (Kotula *et al*, 2014; Mimee *et al*, 2015). Similar to our approach, one group recently administered mCherry-labeled GFP reporter bacteria to

    

germ-free mice and used fluorescence microscopy to measure GFP, and thus gut metabolite levels (Pickard *et al*, 2014). However, due to the relatively small numbers of bacteria that can be analyzed via microscopy, this method is not likely to be extensible to experiments involving an intact microbiota. On the other hand, our approach is compatible with the native gut microbiota, increasing its physiological relevance.

Potential drawbacks of our method include the requirement for flow cytometry equipment to measure fluorescence and the maturation time required for chromophore formation. We show that sfGFP and mCherry maturation is complete after 1 h in the presence of $O_2$ and stable for a minimum of 2 h at 37°C in the presence of a translation inhibitor (Appendix Fig S20). However, other reporter genes enabling colorimetric or luminescence assays could also be used if desired, by adapting previous protocols (Drouault *et al*, 2002; Kotula *et al*, 2014; Mimee *et al*, 2015). Because of the short incubation time (6 h) and presence of the native microbiota, our sensor bacteria likely do not colonize the epithelial mucosal boundary. Thus, sfGFP output from our sensor bacteria likely reflects the luminal concentration of the target metabolite. Finally, the correspondence between sfGFP fluorescence in fecal and colon samples suggests that our method can be used for non-invasive analysis of metabolites in the colon lumen.

The mouse DSS model is one of the most widely used colitis models because of its ease of use and similarity to human ulcerative colitis symptoms (Chassaing *et al*, 2014). DSS administration causes significant inflammation of the large intestine (Okayasu *et al*, 1990) and major disruption of the mucus layer protecting the epithelial lining from pathogen invasion and pro-inflammatory metabolites (Johansson *et al*, 2010, 2014). Increased accessibility to the inner mucus could allow gut bacteria access to elevated levels of the heavily glycosylated, sulfated, and cysteine-rich mucin proteins that are the predominant component of intestinal mucus (Johansson *et al*, 2011). Gut bacteria have evolved the ability to desulfate complex dietary and host polysaccharides to facilitate glycan metabolism (Benjdia *et al*, 2011), which provides liberated sulfate for other gut bacteria to exploit (Rey *et al*, 2013) and has been implicated in colitis (Hickey *et al*, 2015). Both cysteine and sulfate from host mucins can be metabolized to $H_2S$, which is rapidly converted to thiosulfate via enzymatic detoxification in epithelial cells and red blood cells that enter the colon during ulceration. It is worth noting that though DSS has been shown to be resistant to degradation by mouse cecal contents (Kitajima *et al*, 2002), it is possible that some members of the microbiota have the potential to desulfate and/or metabolize DSS, similar to what has been observed for other better studied glycans. We hypothesize that ThsSR is activated by DSS-induced inflammation due to increased gut thiosulfate levels arising via $H_2S$ detoxification, either as a result of mucin degradation or DSS metabolism. If increased $H_2S$ burden is involved in gut inflammation pathogenesis, thiosulfate could serve as a general biomarker beyond the DSS model. Future studies of sulfur metabolism and its role in colitis pathology and mouse gut inflammation models will be enlightening.

Tetrathionate has previously been shown to be elevated in the colonic mucosa of mice infected with a tetrathionate reductase-deficient *S. typhimurium* strain, an alternative inflammation model (Winter *et al*, 2010). In this study, tetrathionate was generated in a ROS-dependent process during inflammation, likely by oxidation of

thiosulfate present in the gut. Given the increased thiosulfate measured in our experiments, we also expected to detect elevated tetrathionate in inflamed animals. However, tetrathionate may be rapidly consumed by the microbiota at the mucosal site of production, resulting in low luminal levels. Protein engineering or genetic memory circuits (Kotula *et al*, 2014; Mimee *et al*, 2015) could be used to increase *S. baltica* TtrSR sensitivity, which could enable detection of lower tetrathionate levels using our method. Additionally, modifications to our protocol enabling the analysis of sensor bacteria that have colonized near the epithelial wall may provide a better readout of tetrathionate concentrations produced during inflammation. Alternatively, it is possible that tetrathionate levels are simply not increased in the DSS model.

Our work demonstrates that engineered sensor bacteria designed to sense and respond to gut metabolites can be used to non-invasively detect colonic inflammation in living mammals. When combined with altered diets (e.g., low sulfur), other inflammation models, more detailed time-resolved assays, *in vivo* imaging methods (Contag *et al*, 1995), or fluorescence microscopy of tissue samples from sacrificed animals (Earle *et al*, 2015; Geva-Zatorsky *et al*, 2015), our sensors could be used to study gut sulfur metabolism and disease with unprecedented resolution. TCSs that sense TMAO (Baraquet *et al*, 2006), nitrate (Rabin & Stewart, 1993), and other TEAs linked to inflammation (Winter *et al*, 2013a) could also be used to study the dynamics of those compounds. Non-TCS sensors that detect inflammation linked compounds such as nitric oxide could also be used (Archer *et al*, 2012). Furthermore, sfGFP could be replaced with colorimetric reporter genes to engineer inexpensive, non-invasive diagnostics, or anti-inflammatory genes to develop "synthetic probiotics" with tissue-specific therapeutic activity (Tabor & Ellington, 2003; Sonnenburg & Fischbach, 2011; Holmes *et al*, 2012; Tabor, 2012). This work also demonstrates that TCS sensors can be mined from genome databases, characterized, and functionally expressed in heterologous bacterial hosts. Many of the thousands of currently uncharacterized TCSs, likely responsive to molecules that span great chemical diversity, could likewise be harnessed for sensing applications in bacterial hosts suited to a wide variety of environments.

## Materials and Methods

### Bioinformatics analysis

A full alignment based on the phosphonate-bd Pfam family (PF12974) was downloaded from the Pfam website and an HMM search (Finn *et al*, 2011) was performed (4/2014) using the alignment queried against the UniProtKB database with default settings. Results were filtered to include proteins containing the domain ordering characteristic of TtrS from *S. typhimurium*: a "phosphonate-bd" ligand-binding domain, followed by a "HisKA" histidine kinase A phosphoacceptor/dimerization domain, and a histidine kinase-like ATPase domain "HATPase_c". The results were downloaded and run through the USEARCH sequence analysis algorithm (Edgar, 2010) to cluster all proteins with > 70% amino acid sequence identity. A centroid representative from each cluster was retrieved and manually examined for proximity to a predicted response regulator and a tetrathionate or thiosulfate utilization

gene. From this list, one putative thiosulfate and tetrathionate candidate was chosen for gene synthesis and validation based on the characteristics described in the results.

## Molecular biology

The *shal_3128*, *shal_3129*, *sbal195_3858*, and *sbal195_3859* genes were codon-optimized for expression in *E. coli* and synthesized by IDT. The inducible RR plasmids were created by cloning the synthesized RRs, Shal_3129 and Sbal195_3858, under the $P_{LTetO-1}$ promoter on a ColE1 backbone with chloramphenicol resistance and constitutively expressed TetR. The full intergenic region upstream of the thiosulfate and tetrathionate reductases ($P_{phsA342}$ and $P_{ttrB344}$) was synthesized by IDT as the output promoters and was cloned upstream of *sfgfp* with BBa_B0034 as the RBS. The inducible SK plasmids were created by cloning the synthesized SKs, Shal_3128 and Sbal195_3859, under the $P_{tac}$ promoter on a p15A backbone with spectinomycin resistance and constitutively expressed LacI. Cloning was performed in NEB-10β cells and sequence-verified plasmids were transformed into BW28357 (CGSC, Yale University) for *in vitro* aerobic characterization experiments.

Constitutive plasmids were created by removing the LacI and TetR cassettes and replacing the inducible promoters with constitutive promoters (Anderson promoter collection: http://parts.igem.org/Promoters/Catalog/Anderson) and designed RBSs (Farasat *et al*, 2014) for fine-tuning of protein expression. Promoters were selected to cover a wide range of predicted strengths. A strong constitutive mCherry marker was incorporated into the RR plasmids to allow for detection of our sensor bacteria from mouse samples. Sequence-verified plasmids were transformed into *E. coli* Nissle 1917 for use in *in vitro* anaerobic and mouse experiments.

All plasmids, truncations, and mutations were constructed using the Golden Gate cloning method (Engler *et al*, 2008). Freezer stocks of plasmid strains were prepared by growing a colony containing sequence-verified plasmid(s) in LB and the appropriate antibiotics (35 µg/ml chloramphenicol and/or 100 µg/ml spectinomycin) to $OD_{600}$ = ~0.5, adding glycerol to a 15% v/v final concentration, and freezing at −80°C.

## *In vitro* aerobic experiments

Overnight cultures were started from freezer stocks in LB with the appropriate antibiotics. 50 µl of overnight culture was added to 3 ml M9 + glycerol (1× M9 salts, 0.4% v/v glycerol, 0.2% casamino acids, 2 mM $MgSO_4$, and 100 µM $CaCl_2$) and grown shaking at 37°C. All characterization experiments were performed aerobically. After 3 h, the cells were diluted to $OD_{600}$ = $10^{-4}$ in 3 ml M9 + glycerol + antibiotics, inducers and ligands [potassium tetrathionate (Sigma-Aldrich) or sodium thiosulfate heptahydrate (Sigma-Aldrich)] were added, and cells were grown for ~6 h to exponential phase ($OD_{600}$ < 0.3). No aTc was required for optimal induction of either sensor. 75 µM IPTG was used for strains harboring pKD182 and 10 µM IPTG was used for strains with pKD227. Culture tubes were then removed from the incubator and placed in an ice water bath to stop growth. 50 µl of cell culture was added to 1 ml ice-cold PBS for flow cytometry analysis. All reported ThsSR and TtrSR with $P_{ttrB344}$ fluorescence values are cellular autofluorescence-subtracted.

## *In vitro* anaerobic experiments

Freezer aliquots of exponentially growing cells were prepared by first diluting 100 µl of an overnight culture grown in M9 + 0.4% glycerol + antibiotics into 3 ml fresh media. After 3 h, the cells were diluted to $OD_{600}$ = $10^{-3}$ in 3 ml M9 + 0.4% glycerol + antibiotics and cells were grown to $OD_{600}$ = ~0.133. Cells were mixed with filter-sterilized glycerol to a final concentration of 15% v/v glycerol and a final $OD_{600}$ = 0.1, aliquoted into single use vials, and frozen at −80°C.

M9 media (no glycerol or antibiotics) was equilibrated in an anaerobic chamber overnight prior to experiments. Freezer aliquot cells were added to anaerobic media to a final concentration of 0.4% glycerol (1:37.5 dilution with an initial $OD_{600}$ = $2.67 × 10^{-3}$) along with ligand. Cells were grown in an anaerobic chamber for 6 h at 37°C and placed in an ice water bath when finished. 50 µl of cells was added to 500 µl PBS + 1 mg/ml chloramphenicol to halt protein translation. Cells were incubated in a 37°C water bath for 1 h to allow maturation of sfGFP and mCherry fluorophores. Chloramphenicol resistance is encoded on the RR plasmid; however, 28.5-fold excess antibiotic was used relative to the plasmid maintenance concentration, which should be sufficient to overcome inactivation. Previous antibiotic screens using similar plasmid backbones identified chloramphenicol as the best performing translation inhibitor (EJ Olson, unpublished data). Additionally, time course experiments in these strains show that sfGFP fluorescence reaches a maximum at 1 h and is stable for up to 2 h when incubated with chloramphenicol but not without it (Appendix Fig S19). After fluorophore maturation, cells were placed in an ice water bath and analyzed by flow cytometry. Reported fluorescence values are not corrected for cellular autofluorescence.

## *Ex vivo* colon experiment

Whole colons were removed from healthy C57Bl/6 mice and tied at the ends with 5-0 Vicryl suture string (Ethicon, Somerville, NJ, USA). Fecal pellets were left in the colon intact in order to allow for a "native" environment for the sensor bacteria and ligands to interact. The colonic loops were submerged in anaerobically pre-reduced DMEM (Life Technologies, Grand Island, NY, USA) with 10% fetal bovine serum (FBS). For the analysis of the sensor, concentrations of the ligands representing saturating concentrations (1 mM tetrathionate and 5 mM thiosulfate) were added to the media, and then 100 µl of the ligands in media was injected into the luminal space of the colon. In addition, 100 µl of the sensing bacteria was also injected into the luminal space, for a total of $10^9$ colony-forming units (CFUs). The colons were incubated at 37°C for 6 h under anaerobic conditions, and then all external media and internal fecal slurry were collected on ice separately for analysis via flow cytometry for total GFP output.

## Dextran sodium sulfate mouse experiments

Six- to eight-week-old male C57BL/6 mice were procured from the Center for Comparative Medicine (CCM) Production Colony at the Baylor College of Medicine in Houston, Texas. Mice were transferred to an established protocol that was approved by the Baylor College of Medicine Institutional Animal Care and Use

Committee (IACUC). DSS-exposed mice were given 3% (w/v) DSS (MW = 36–50,000; MP Biomedicals) in drinking water *ad libitum*, and control mice were given untreated drinking water *ad libitum*. Mice were randomized into group according to co-housing within cages, by randomly selecting each cage for DSS treatment or control. Standard rodent diet (5V5R/PicoLab Select rodent Diet 50IF/6F, Labdiet; > 17% protein, sulfur content = 0.21%) was provided *ad libitum* over the course of the study. Mice were treated with DSS for 5 days. On the final day of DSS treatment, fecal pellets were collected from all DSS and control mice. These mice were then orally gavaged with either $10^9$ CFUs of *E. coli* Nissle containing the sensor or $10^9$ CFUs of *E. coli* Nissle containing a sensor with the RR inactivating mutation (D57A for thiosulfate and D55A for tetrathionate). Mice were randomly matched with sensors. 6 h after sensor gavage, fecal pellets were collected from all mice for GFP analysis via flow cytometry. Next, the mice were humanely euthanized and luminal contents from the proximal and distal portions of the colon were collected for sfGFP analysis. Distal and rectal sections of the colon tissue were fixed in 10% neutral-buffered formalin for 24 h before transfer to 70% ethanol. These tissues were paraffin-embedded and hematoxylin and eosin (H&E) staining was performed for colitis scoring by the Texas Medical Center Digestive Diseases Center. Blinded histologic scoring was performed using previously described methods (Chassaing *et al*, 2014). Briefly, a value is assigned from 1 (moderate) to 3 (severe) to evaluate each one of the following features: the extent of epithelial damage and the inflammatory infiltration in the mucosa, submucosa, and muscularis/serosa. The number obtained for each characteristic was multiply by 1 (focal), 2 (patchy), or 3 (diffuse), depending on the lesion extension, resulting in an overall score from 0 (no inflammation) to 36 (maximal inflammation).

### Colon and fecal sample preparation

Contents of the proximal and distal colon and fecal samples, if available, were homogenized in 1 ml of PBS + 1 mg/ml chloramphenicol using a pipet tip. Samples were vortexed for 1 min and filtered through a 5-μm syringe filter (Pall Laboratory, VWR catalog number 28150-956) to remove solids and murine cells but allow passage of bacterial cells. An additional 1 ml of PBS + chloramphenicol was washed through the syringe filter to extract bacteria from the hold-up volume. Filtered samples were incubated for 1 h in a 37°C water bath to allow for maturation of fluorophores and were transferred to a 4°C refrigerator. Samples were analyzed by flow cytometry less than 24 h after sample collection.

The Shapiro–Wilk test for normality was administered to the data for each DSS-treated and water control comparison. Any comparison for a non-normal distribution was made with the non-parametric Mann–Whitney *U*-test. For normal distributions, equality of variance was confirmed via Levene's test and then directly compared with *t*-tests. All statistical testing was performed using R 3.2.3, in RStudio (R Core Team, 2015).

### Flow cytometry and data analysis

Flow cytometry analysis was performed on a BD FACScan flow cytometer with a blue (488 nm, 30 mW) and yellow (561 nm, 50 mW) laser. Fluorescence was measured on three channels: FL1 with a 510/20-nm emission filter (GFP), FL2 with a 585/42-nm filter (GFP/mCherry), and FL3 with a 650-nm long-pass filter (mCherry).

For pure *E. coli* culture experiments, cells were thresholded by an SSC scatter profile characteristic of the strain used. Typical event rates were between 1,000 and 2,000 events per second for a total of 30,000 events. Mouse colon and fecal samples were both thresholded in the FL3 channel, to ignore counts with low mCherry-like fluorescence, and gated by an FSC/SSC scatter characteristic of *E. coli* Nissle. Data were collected for 5 min or for 30,000 counts within the gated population, whichever came first. Calibration particles (Spherotech, catalog RCP-30-20A) were run at the end of every experiment at the gain settings used for data collection.

After data acquisition, raw data were processed using FlowCal (Castillo-Hair *et al*, 2016). First, a standard curve was created from the calibration beads to convert arbitrary units into absolute fluorescence units (MEFL for FL1 and MECY for FL3). Second, data were gated by an FSC/SSC scatter profile characteristic of *E. coli* Nissle and by FL2 and FL3 fluorescence values, discarding counts with an FL2 value lower than 250 a.u. and an FL3 value lower than 9,000 MECY. Samples giving fewer than 250 counts by these standards were discarded. Overall, DSS-treated mice gave more counts/sample and usable samples than untreated mice.

### Hill function fitting

The transfer functions were obtained by fitting the averaged fluorescence values at each ligand concentration to the Hill equation, $F = A + B/(1 + (k_{1/2}/L)^n)$, where F is the fluorescence at a given ligand concentration L, $k_{1/2}$ is the concentration of agonist that elicits a half-maximal response, $n$ is the Hill coefficient, A is the fit of the minimum response with no ligand, and B is the fit of the maximum fluorescence response at saturating ligand concentration.

**Expanded View** for this article is available online.

### Acknowledgements

We thank Sebastian Winter for the kind gift of *E. coli* Nissle 1917, Brian Landry for help with developing the flow cytometry protocol of mouse colon/fecal samples, Kathryn Brink for assistance with the ROC analysis, and Nicholas Ong for assistance with flow cytometry data visualization. We would also like to thank Dr. Joel Moake and his lab for use of the flow cytometer. This work was supported by the Welch Foundation (C-1856), an ONR Young Investigator Award (N00014-14-1-0487), and an NSF CAREER Award (1553317) to JJT, an R01 grant (CA1428260) to N.F.S, and seed funds from Baylor College of Medicine to R.A.B. K.N.D. was supported by a Rice Department of Bioengineering Postdoctoral Fellowship.

### Author contributions

KN-MD and JJT conceived of the project. KN-MD and RUS performed bioinformatics analysis to identify sensors. RUS optimized mCherry expression for *in vivo* experiments. KN-MD built and characterized sensors *in vitro*. KN-MD, JDG, LCO-V, and NFS performed *in vivo* experiments. LCO-V and COB performed histopathology analysis. KN-MD, JDG, LCO-V, NFS, RAB, and JJT designed experiments and analyzed results. KN-MD and JJT wrote the manuscript with feedback from all authors.

## Conflict of interest

Rice University has filed for a patent covering the use of ThsSR as a biosensor to diagnose or treat gut inflammation.

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
