## [Review Process File · Molecular Systems Biology]

Engineering bacterial thiosulfate and tetrathionate sensors for detecting gut inflammation

Kristina Daeffler, Jeffrey Galley, Mr. Ravi Sheth, Ms. Laura Ortiz-Velez, Christopher Bibb, Noah Shroyer, Robert Britton and Jeffrey Tabor

Corresponding author: Jeffrey Tabor, Rice University

Review timeline:

Submission date:	28 October 2016
Editorial Decision:	13 December 2016
Revision received:	14 February 2017
Editorial Decision:	10 March 2017
Revision received:	14 March 2017
Accepted:	15 March 2017

Editor: Maria Polychronidou

Transaction Report:

1st Editorial Decision

13 December 2016

Thank you again for submitting your work to Molecular Systems Biology. We have now heard back from the three referees who agreed to evaluate your study. As you will see below, the reviewers acknowledge that the presented sensors are a useful tool for future analyses. They raise however a series of concerns, which we would ask you to address in a revision.

The reviewers' recommendations are rather clear so there is no need to repeat the points listed below. One of the more fundamental issues refers to the need to include additional analyses to provide better support for the function of the sensors in vivo.

REFeree REPORTS

Reviewer #1:

The manuscript by Daeffler et al. describes bacterial thiosulfate and tetrathionate sensors for the detection of gut inflammation.

The paper comprises two parts: first, identification, reconstitution in *E. Coli* Nissle, and testing of the aforementioned two-component signaling pathways. This part is well-done, detailed and thorough. The results will be of interest to a wide community of microbiologists and synthetic biologists. I do not have criticisms here.

In the second part, the engineered bacteria are used as inflammation sensors in mice. I am afraid that

this second part is lacking. The most striking result is a very weak correlation between histological inflammation markers and the sensor response.

The histological scoring is not a "gold standard", and I am surprised the authors did not use multiple pathologists to score the same samples to increase their confidence. However, this is not the point. The bacteria should detect their respective ligands, thiosulfate and tetrathionate, in the gut. To prove this statement, there must be an independent, direct, chemically-based analytical measurement of these species in the relevant physiological samples. For example, this method might be used <https://www.ncbi.nlm.nih.gov/pubmed/19559870> or another similar method.

Granted, the authors performed extensive titrations and measurements in vitro. However, the responses in vivo are something entirely different, as the bacteria might lose their payload (even if the residence time is short), etc. Thus having an independent measurement of the ligand and the correlation between this measurement and the biosensor output is crucial to support the conclusion of this study. The fact that the correlation between inflammation and sensor output is weak, should be worrying.

Minor comments

- 1) I request that 2D flow cytometry data (Cherry vs GFP) of all the animal-derived bacterial samples be shown in supplementary information.
- 2) ROC curves should be built to illustrate sensor diagnostic performance; AuROC should be calculated and reported.

Reviewer #2:

The authors used a bioinformatics approach to identify two component systems from *Shewanella* bacteria capable to sensing thiosulfate and tetrathionate. The authors then implemented these systems first in laboratory *E. coli* and find that they are functional. The authors go on to implement these systems in *E. coli* Nissle probiotic bacteria and show that they are again functional. They then show that when these engineered bacteria are administered to colitis model mice by oral gavage, they provide a readout of gut thionate levels and that these readouts correlate with gut inflammation by histology scoring. In general, this manuscript is easy to read, provides compelling arguments, and presents well-supported data.

I believe this work to be significant. To my knowledge, this work is the first example of an engineered bacteria with a thiosulfate sensor that is functional in a mouse model. This builds off previous work. Previous work provided the mouse model for colitis and *E. coli* Nissle bacterial chassis strain; evidence of correlation of thiosulfate and tetrathionate to gut inflammation; biochemical understandings of thiosulfate, tetrathionate, and two component systems in general; and a previously characterized thiosulfate sensory system to which the current system can be compared. While building off previous work, these authors put the whole system together, with new sensors, for the first time. By providing new and useful thiosulfate and tetrathionate sensors to the community, this work provides methods for implementing and analyzing engineered probiotic systems as biomarker readouts, and lays the groundwork for the system that may be able to detect gut inflammation in humans. The work should be of interest to a large audience of researchers including: synthetic biologists, gastroenterologists, gut microbiologists, and bioinformaticians. I think it is well suited for *Molecular Systems Biology*, but believe there are some points to be addressed before suitable for publication.

1. In figure 2D, what are the range of concentrations used to examine specificity. I believe I missed it, but clarifying physiological ranges in the text would strengthen the authors' results.
2. It might be beneficial to expose Nissle carrying these sensors to DSS outside of the host organism to determine if they show differences in sfGFP and mCherry response due to this molecule being present in its environment alone. I am wondering if this treatment somehow impacts the sensor.

3. In the comparisons of this work's FACS method, it would be fair to discuss the possible drawbacks of this method such as the maturation time required and the requirement of FACS equipment to provide accurate readouts. The authors could comment on how these issues could be overcome (such as by the use of colorimetric indicators as explained later in the discussion).

4. Judging by the data in panel F of Figure 5 the claim that "in vivo activity of our thiosulfate sensor is proportional to gut inflammation" found on pages 17 and 6 might be better put "in vivo activity of our thiosulfate sensor correlates with gut inflammation." Correlation is clear,, though weak. I find proportionality as too strong of wording and believe this to be one of the weaker conclusions in the paper. Yet, this is an important conclusion.

5. When the authors state that the ThsSR strain operates robustly, I feel this may be a slight overstatement. I feel the data show it works in most instances, but not all.

6. Is there a way for the authors to figure out if the TtrSR strain circuitry is broken, or the concentration of tetrathionate is low? Is there any way to test for tetrathionate concentration directly? HPLC methods, etc. Although the article did not claim one or the other conclusion, this experiment would help decouple the question of whether the sensor doesn't work in vivo or if levels are too low to detect -- providing valuable insights to the field. I believe this is important.

-Presentation and style

Label plasmid numbers in Figure 2A and 3C

Indicate $k_{1/2}$ and n in Figure 2C and 3G, or at least in caption.

Reviewer #3:

Summary

In this study, Daeffler et al. reported the identification and characterization of one thiosulfate and one tetrathionate sensors from marine *Shewanella*. The two sensing systems were reconstituted in *E. coli* and different substrates were tested to show the efficiency and high substrate specificity for these two systems when tested in tube. The authors further optimized the sensing system and gavaged the engineered strains into mice. The authors show that the activity of the thiosulfate sensor is proportional to gut inflammation claiming that this molecule could be a novel inflammation biomarker. In comparison, tetrathionate sensor shows low in vivo activity at high inflammation levels. Comparing to previous literature, the highlights of the paper lie in that 1. the identification and characterization of a new thiosulfate sensor, 2. the authors have shown that the thiosulfate sensing system works well, non-invasively, in both health and DSS mouse model as a reporting tool for gut inflammation. In general, the publication in *Molecular System Biology* is recommended after the comments below have been addressed.

Major points

1. The experiments reported for characterization of both thiosulfate and tetrathionate-sensing TCSs are somewhat similar to each other. It will be better if the authors could combine these two parts into one in the results section.

2. The authors show that the expression of sfGFP levels in thiosulfate sensing system are weakly proportional to histologic score. How does sfGFP level relate to the thiosulfate level in the mice gut? It will be more convincing if the authors could first show that the sfGFP expression level is in good correlation with the concentration of thiosulfate molecules in the (healthy or DSS) mice gut, then shows that it is proportional to inflammation level. This will bridge the gap between in vitro aerobic test of their sensing system and the in vivo outputs (sfGFP expression levels). This will also set up a direct linkage between thiosulfate concentration and inflammation in DSS disease mice (although more experiments are definitely necessary to show the validity of thiosulfate as a biomarker for gut inflammation, these experiments do not have to be included in this study). It is the same for tetrathionate sensing system as well.

3. It might be helpful to normalize the population of the engineered E.coli that has been gavaged with other microbial members in the healthy and DSS disease mice via sequencing. This will show how much E. coli is actually there after 6hrs gavage and euthanization. One reason is because the microbiota sometimes changed a lot in diseased DSS model. An elevated population of E coli in the DSS mice would also lead to the upregulation of sfGFP levels comparing to those healthy subjects, in which the gavaged strain might be cleared off much faster. This might help to explain why the sensing system for tetrathionate barely works in vivo because one of the possibility is the strain gets rapidly cleaned off the gut in DSS mice comparing to the healthy group.

Minor points

1. I would recommend combining Figures 2 and 3 into one figure.

1st Revision - authors' response

14 February 2017

Response to Reviewers

We thank the three Reviewers for reading our manuscript carefully and providing helpful feedback. We respond to each Reviewer criticism below.

Reviewer #1:

...In the second part, the engineered bacteria are used as inflammation sensors in mice. I am afraid that this second part is lacking. The most striking result is a very weak correlation between histological inflammation markers and the sensor response.

We observe a statistically significant increase in ThsSR output when mice have inflammation, with zero false positives (**Fig. 5B**). This response is completely abolished in a negative control experiment where ThsSR is disabled by mutation (**Fig. 5C**). These results demonstrate that we can use our engineered thiosulfate-sensing bacteria to detect a diseased state *in vivo*. We believe this is a significant advance. Due to the complexity of the gut environment, the fact that we measure our bacteria at a single time point, and the subjective nature of pathological scoring, we do not find it too surprising that the correlation between inflammation score and sfGFP levels is weak. Future advances in our sensor or strain design, *in vivo* protocol, or analytical methods, particularly those enabling increased spatial and temporal resolution *in vivo*, all may improve the strength of the correlation between inflammation and ThsSR output. Nonetheless, we have removed the corresponding figure and all claims of proportionality between ThsSR activity and extent of inflammation from the manuscript.

The histological scoring is not a "gold standard", and I am surprised the authors did not use multiple pathologists to score the same samples to increase their confidence.

We thank the Reviewer for this feedback. We have increased confidence in our inflammation scoring by having a second blinded pathologist score the samples (Appendix **Fig. S24**), and described this change in the main text.

However, this is not the point. The bacteria should detect their respective ligands, thiosulfate and tetrathionate, in the gut. To prove this statement, there must be an independent, direct, chemically-based analytical measurement of these species in the relevant physiological samples. For example, this method might be used <https://www.ncbi.nlm.nih.gov/pubmed/19559870> or another similar method.

Accurate measurement of many metabolites from animal stool by direct chemical methods is highly challenging. Thiosulfate and tetrathionate are colorless molecules and therefore need to undergo a chemical transformation to be easily distinguished by chromatography, requiring high specificity of reactivity toward the molecule of interest over other molecules in the sample. The method that the Reviewer provides in the above reference is used for urine, which is low in sulfide and other sulfur-donors that would react with their derivatizing agent, thereby enabling their successful quantification. However, the chemical reaction they use is not specific for thiosulfate over other

sulfur-donors likely to be found in fecal samples (sulfide, iron-sulfur clusters, etc.). While it is likely possible with the right columns, running buffers, and analytical equipment to develop a protocol to quantify these molecules, it would require significant development, be very costly, and is outside the scope of this study.

Commercial analysis is another option. However, Metabolon, a leading company in this space, quoted us \$17,000 and a four-month turn-around for qualitative analysis of thiosulfate with an unknown lower limit of detection. Furthermore, they cannot measure tetrathionate. Thus, we aim to develop our own quantitative thiosulfate and tetrathionate analytical methods in future work.

Nonetheless, we clearly demonstrate that our engineered bacteria detect and report inflammation, as discussed above. We believe this is an important and non-trivial advance and sufficient for this first report. To address the fact that we do not perform analytical chemical measurements, we have removed all claims that we are directly detecting thiosulfate in the gut. Finally, as we write in the Discussion, our study is the first to *suggest* thiosulfate may be an inflammation biomarker and will motivate future studies into gut thiosulfate levels, gut sulfur metabolism, and inflammation.

Granted, the authors performed extensive titrations and measurements in vitro. However, the responses in vivo are something entirely different, as the bacteria might lose their payload (even if the residence time is short), etc. Thus having an independent measurement of the ligand and the correlation between this measurement and the biosensor output is crucial to support the conclusion of this study. The fact that the correlation between inflammation and sensor output is weak, should be worrying.

We completely agree with the Reviewer that *in vitro* and *in vivo* environments are very different. To address this concern, we have performed additional experiments wherein we harvest colons from healthy mice, inject our sensor bacteria and the relevant ligand (thiosulfate or tetrathionate), incubate for 6 hours, harvest the colon contents, and measure the response of our sensors by flow cytometry (Appendix Fig. S21 and S22). This experiment shows that our bacteria do not lose their payloads (i.e. sensors) after a 6-hour incubation in the colon without antibiotics and that both of our sensors are activated by their respective ligands in the complex colon environment. This *ex vivo* experiment provides strong additional evidence that our sensors function as designed *in vivo*.

Minor comments

1) I request that 2D flow cytometry data (Cherry vs GFP) of all the animal-derived bacterial samples be shown in supplementary information.

We have added all data in Appendix Fig. S27.

2) ROC curves should be built to illustrate sensor diagnostic performance; AuROC should be calculated and reported.

We have built ROC curves and calculated AuROC to be 0.8692 for ThsSR *in vivo* due to the strong response to inflammation and very low false positive rate (Appendix Fig. S25).

Reviewer #2:

1. In figure 2D, what are the range of concentrations used to examine specificity. I believe I missed it, but clarifying physiological ranges in the text would strengthen the authors' results.

We tested 10 mM of each ligand. We have incorporated this information into the text and the Fig. 2D legend for clarity.

2. It might be beneficial to expose Nissle carrying these sensors to DSS outside of the host organism to determine if they show differences in sfGFP and mCherry response due to this molecule being present in its environment alone. I am wondering if this treatment somehow impacts the sensor.

We agree that this is an important experiment. As shown in Appendix Fig. S23, DSS does not have any effect on either of our sensors *in vitro*.

3. In the comparisons of this work's FACS method, it would be fair to discuss the possible drawbacks of this method such as the maturation time required and the requirement of FACS equipment to provide accurate readouts. The authors could comment on how these issues could be overcome (such as by the use of colorimetric indicators as explained later in the discussion).

This is an interesting point and we have now added the requested discussion in the text:

“Potential drawbacks of this method include the requirement for flow cytometry equipment to measure fluorescence and the maturation time required for chromophore formation. We show that sfGFP and mCherry maturation is complete after one hour in the presence of oxygen and stable for a minimum of two hours at 37C in the presence of a translation inhibitor (Appendix Fig. S19), however other readouts including colorimetric assays could be used if necessary.”

4. Judging by the data in panel F of Figure 5 the claim that "in vivo activity of our thiosulfate sensor is proportional to gut inflammation" found on pages 17 and 6 might be better put "in vivo activity of our thiosulfate sensor correlates with gut inflammation." Correlation is clear,, though weak. I find proportionality as too strong of wording and believe this to be one of the weaker conclusions in the paper. Yet, this is an important conclusion.

We agree that may have been too strong of a claim and have removed the sfGFP vs. histopathology score panels from Fig. 5 and the corresponding text. Instead we give the ROC curve requested by Reviewer #1 as a better indicator of sensor performance as a diagnostic of gut inflammation.

5. When the authors state that the ThsSR strain operates robustly, I feel this may be a slight overstatement. I feel the data show it works in most instances, but not all.

We thank the Reviewer for noticing this issue. We have changed the statement to say: “These results demonstrate that ThsSR can be activated in a living mouse gut and indicate that thiosulfate may be elevated upon DSS treatment.”, which we believe is supported by the data.

6. Is there a way for the authors to figure out if the TtrSR strain circuitry is broken, or the concentration of tetrathionate is low? Is there any way to test for tetrathionate concentration directly? HPLC methods, etc. Although the article did not claim one or the other conclusion, this experiment would help decouple the question of whether the sensor doesn't work in vivo or if levels are too low to detect -- providing valuable insights to the field. I believe this is important.

Our newly added colon explant experiments show that the sensor is still functional after 6 hours of incubation in the colon environment. We agree that direct quantification of thiosulfate and tetrathionate would further strengthen our conclusions. However, as mentioned above, direct measurement of thiosulfate and tetrathionate in the gut is outside the scope of this study.

-Presentation and style

Label plasmid numbers in Figure 2A and 3C

Plasmid numbers have been labeled in the corresponding figure panel.

Indicate $k_{1/2}$ and n in Figure 2C and 3G, or at least in caption.

$k_{1/2}$ and n have been labeled in the corresponding figure panel.

Reviewer #3:

1. The experiments reported for characterization of both thiosulfate and tetrathionate-sensing TCSs are somewhat similar to each other. It will be better if the authors could combine these two parts into one in the results section.

We thank the Reviewer for this suggestion. Indeed, we originally wrote the manuscript in this way. However, we ultimately decided to separate the two sensor characterization sections in the text. Although the experimental methods are similar, there are considerable differences between the two sensors, which became confusing to discuss together. However, in response to the Reviewer's comment, we have shortened the description of TtrSR characterization to minimize redundant explanations.

2. The authors show that the expression of sfGFP levels in thiosulfate sensing system are weakly proportional to histologic score. How does sfGFP level relate to the thiosulfate level in the mice gut? It will be more convincing if the authors could first show that the sfGFP expression level is in good correlation with the concentration of thiosulfate molecules in the (healthy or DSS) mice gut, then shows that it is proportional to inflammation level. This will bridge the gap between *in vitro* aerobic test of their sensing system and the *in vivo* outputs (sfGFP expression levels). This will also set up a direct linkage between thiosulfate concentration and inflammation in DSS disease mice (although more experiments are definitely necessary to show the validity of thiosulfate as a biomarker for gut inflammation, these experiments do not have to be included in this study). It is the same for tetrathionate sensing system as well.

We agree that directly correlating thiosulfate levels with GFP output would strengthen our interpretations of *in vivo* sensor performance. However, as discussed in the response to Reviewer #1, developing a method for this is difficult. We have added the colon explant experiment and ROC analysis to strengthen our interpretations. We have edited the manuscript to avoid conclusions involving *in vivo* thiosulfate concentrations.

3. It might be helpful to normalize the population of the engineered E.coli that has been gavaged with other microbial members in the healthy and DSS disease mice via sequencing. This will show how much E. coli is actually there after 6hrs gavage and euthanization. One reason is because the microbiota sometimes changed a lot in diseased DSS model. An elevated population of E coli in the DSS mice would also lead to the upregulation of sfGFP levels comparing to those healthy subjects, in which the gavaged strain might be cleared off much faster. This might help to explain why the sensing system for tetrathionate barely works *in vivo* because one of the possibility is the strain gets rapidly cleaned off the gut in DSS mice comparing to the healthy group.

Using flow cytometry we measure whole cell fluorescence from >250 individual bacterial counts and can distinguish our sensor bacteria with strong constitutive mCherry expression from other bacteria in the gut. The sfGFP values reported are the geometric mean of individual cell fluorescence measurements and do not reflect bulk GFP, which would require normalization to cell count. We do see different bacterial counts at the end of the 6 hour experiment between healthy and DSS-treated mice, with higher counts for DSS-treated mice. Enterobacteria blooms have been observed during inflammation, but we don't believe that this will have an effect on sfGFP levels.

Minor points

1. I would recommend combining Figures 2 and 3 into one figure.

We thank the reviewer for the suggestion. In an initial version of the manuscript, we did combine Figures 2 and 3. However, as described above, we believe that this reduces readability because the systems have many different details. Thus, we have chosen to leave them separate.

2nd Editorial Decision

10 March 2017

Thank you for sending us your revised manuscript. We have now heard back from reviewer #2 who agreed to evaluate the study. As you will see below, s/he thinks that the major issues have been satisfactorily addressed. However, s/he raises a couple of remaining concerns, which we would ask you to address in a minor revision.

REFEREE REPORT

Reviewer #2:

I thank the authors for their work in carefully addressing the reviewer comments. I still believe that this manuscript is well suited for Molecular Systems Biology because the engineered thiosulfate-sensing bacteria detect a diseased state *in vivo* - even though further optimization could improve the system. After considering the revisions, the following points remain:

1. I would still like to see the plot showing GFP readout with histological scoring data included in the supplement. While I do not believe that this data is sufficient to make claims of proportionality, I do believe that it could provide valuable information to the field about the behavior of the sensor and the robustness of the FACS method presented in this work. Including only the ROC curve in S25 seems to obscure this information. Please also clarify if the ROC curve was generated using a "diagnosis" by histological scoring or simply the addition of DSS.

2. I thank the authors for performing ex vivo analysis as shown in S21 and S22. This data brings two points to my attention:

(i) the first is that the D55A mutation seems to only attenuate rather than completely "abolish" the response of the TtrSR both in vivo and in vitro. Am I seeing that correctly? I am not sure as to the significance. If it is attenuated, perhaps the wording should be changed.

(ii) the other is that the authors should use a statistical test to show significance of the increase in fluorescence in the ex vivo WT columns of panel G of S21 and note this result in the figure or in the caption.

In conclusion, I believe that the authors sufficiently addressed the reviewer comments and recommend this article for publication after these minor points are addressed.

2nd Revision - authors' response

14 March 2017

Reviewer #2 comments

1. I would still like to see the plot showing GFP readout with histological scoring data included in the supplement. While I do not believe that this data is sufficient to make claims of proportionality, I do believe that it could provide valuable information to the field about the behavior of the sensor and the robustness of the FACS method presented in this work. Including only the ROC curve in S25 seems to obscure this information.

Thank you. We agree and have now added the GFP fluorescence vs. histology scores to the Appendix (Figure S27).

Please also clarify if the ROC curve was generated using a "diagnosis" by histological scoring or simply the addition of DSS.

We generated the ROC curve based on the addition of DSS. We have clarified this fact in the main text and in the figure caption.

2. (i) The D55A mutation seems to only attenuate rather than completely "abolish" the response of the TtrSR both in vivo and in vitro. If it is attenuated, perhaps the wording should be changed.

The reviewer is correct. We have changed the wording accordingly.

(ii) The other is that the authors should use a statistical test to show significance of the increase in fluorescence in the ex vivo WT columns of panel G of S21 and note this result in the figure or in the caption.

While the flow cytometry histograms clearly show sensor activation in these experiments, we don't believe a statistical test is meaningful for such a small sample size. However, we have now added the p-values for the equal and unequal variance assumptions.

Thank you again for sending us your revised manuscript. We are now satisfied with the modifications made and I am pleased to inform you that your paper has been accepted for publication.

Corresponding Author Name: Jeffrey Tabor

Journal Submitted to: MSB

Manuscript Number: MSB-16-7416